# Mystical Experience: Women's Pathway to Knowledge

**Maria Clara Bingemer** 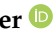

Department of Theology, Pontifical Catholic University of Rio de Janeiro, Rio de Janeiro 22451-900, RJ, Brazil; agape@puc-rio.br

**Abstract:** This article aims to reflect on two issues within the vast area of mystical studies: 1. The link between mystical experience and knowledge; 2. Mystical experiences lived through by women as a pathway to and from knowledge. Firstly, we will try to circumscribe the concept of mystical experience by retrieving some of the main thoughts of scholars who have studied the mystical phenomenon and the writings of individuals who experienced it. Secondly, we will pursue our reflection with the aid of philosophers and theologians who thought and argued that mystical experience is and contains knowledge and bears not only affective and spiritual, but also intellectual, fruit. Thirdly, we will attempt to demonstrate how, throughout the history of Christianity, a significant number of women were the specific protagonists of this synthesis between experience and knowledge and how this allowed them to bring original contributions to their context and historical time. We conclude with a detailed commentary and reflection on the French 20th-century mystic Simone Weil who, as both an intellectual and a mystic, was a pioneer in bringing a prophetic vision on issues that would inspire society, the Church, and spiritual life many decades after her death. Our conclusions will be based on this and on the countercultural benefits that mystical experience lived through by women can bring to contemporary times.

**Keywords:** mysticism; mystical experience; knowledge; women; gender; theology

The mystical experience is generally understood as an affective one, made of love and union. The history of Christianity has been marked by this type of experience and owes to it some of its most luminous milestones and highlights. Christian mystics have been great founders, bright intellectuals, and paradigmatic figures in raising new issues for Theology and Philosophy theology, philosophy, social justice, and politics. In this article we wish to reflect and write about two issues in the vast area of mystical studies, focusing specifically on Christian mysticism:[1] 1. The link between mystical experience and knowledge; 2. Mystical experiences lived through by women as a pathway to and from knowledge. We will briefly highlight a few women mystics in order to set the stage for the topic to be further developed below.[2]

Firstly, we will attempt to circumscribe the concept of mysticism by retrieving some of the main thoughts of scholars who have studied the mystical phenomenon and writings of individuals who experienced it. For this purpose, we will apply some elements from philosophy, but mostly from theology. Secondly, we will pursue our reflection with the aid of thoughts by philosophers and theologians who thought and argued that mystical experience is and contains knowledge and bears not only affective and spiritual, but also intellectual, fruit.[3] Thirdly, we will attempt to show how a significant group of women were the specific protagonists of this synthesis between experience and knowledge and how this allowed them to bring original contributions to their context and historical time.

We conclude with a detailed commentary and reflection on the French 20th-century mystic Simone Weil who, as both an intellectual and a mystic, was a pioneer in bringing a prophetic vision on some issues that would inspire society, the Church, and spiritual life many decades after her death. As such, she became a paradigmatic figure who demonstrated that intellectual ability does not entail only rational thinking, but consists in a great

level of spiritual sensitivity, which brings altogether an enormous responsibility in leading humanity towards fulfilling its vocation to live fully. Our conclusions will be based on the countercultural benefits that mystical experience, as lived through by women, can bring to contemporary times.

## 1. Mystical Experience: Discussions on the Concept

Mysticism and the mystical experience are increasingly becoming an important area of study in the religious sciences. They have attracted attention in different disciplines, such as Literature, Philosophy, Anthropology literature, philosophy, anthropology and the Social Sciences, Psychology social sciences, psychology, and also Theologytheology.[4] One of the central questions of these disciplines is the connection between mysticism and knowledge; in other words, the notion that the mystical experience is an experience of knowledge.

The impulse at the origin of the mystical experience would then be to feel a sense of the Absolute. This would mean, in other words, a desire to feel the Mystery, the Transcendent, the Divine, and to experience union with this Mystery. No effort, no experiment, no ascesis, no self-surrender would be considered excessive and above one's own strength when faced with the perspective of being immersed in the light which clarifies the vision and purifies the heart of one who possesses this true desire (Ames 1915, p. 254). The process of opening and purifying the whole self in order to receive the enlightenment, which will enable one to be guided along the path of union with God, is dynamic and embodies an impulsive quality; it also involves sensuous imagery, and is social in character (Ibid.). The experience to be then encountered and felt will be recognized as knowledge. There is, in the mystical experience, a feeling of transitioning from darkness to light, from ignorance to wisdom, which opens up vision and consciousness to new inspiration and revelations.

The pages and narratives of many mystics are full of this longing and enlightenment. Teresa of Avila, for instance, in her book *Life*, describes how God personified was the master of her life and taught her through her experiences:

> Or it may be, as His Majesty has always been my Master may He be blessed for ever! for I am ashamed of myself that I can say so with truth that it was His good pleasure I should meet with no one to whom I should be indebted in this matter. So, without my wishing or asking it I never was careful about this, for that would have been a virtue in me, but only about vanity God gave me to understand with all distinctness in a moment, and also enabled me to express myself, so that my confessors were astonished; but I more than they, because I knew my own dullness better. It is not long since this happened. And so that which our Lord has not taught me, I seek not to know it, unless it be a matter that touches my conscience (http://www.saintsbooks.net/books/St.%20Teresa%20of%20Avila%20-%20The%20Life%20of%20by%20Herself.pdf accessed on 5 October 2022).

Equally, Ignatius of Loyola describes his experience as one of being taught by God like a pupil by a teacher:

> At that period God dealt with him as a teacher instructing a pupil. Was this on account of his ignorance or dullness, or because he had no one else to teach him? Or on account of the fixed resolve he had of serving God, with which God Himself had inspired him, for the light given him could not possibly be greater? He was firmly convinced, both then and afterward, that God had treated him thus because it was the better spiritual training for him. (O'Conor 2019, p. 16)

Nevertheless, mystical experience also appears as "an experience beyond all experience, a knowledge over all knowledge" (Nef 2018, p. 11). Some scholars and commentators have written recently on this subject, distinguishing between a doctrinal mysticism, which defines and enounces dogmatically on the nature and essence of God and the configuration of the union it may bring, and an experiential mystical theology, carried out mostly by saints and contemplatives; in short, by mystics themselves (Ibid., p. 254).

When speaking about mystical experience as knowledge then, we must attempt to circumscribe the concept. Perhaps few concepts in philosophy and theology are as difficult to demarcate through a precise and adequate definition as that of "mysticism". Several authors and various currents have sought to define it without reaching a definitive agreement on the subject. Here we will examine the contributions of a few 19th and 20th century thinkers on the topic.[5]

In philosophy, William James' book *Varieties of Religious Experience* (James 1929) is a reference. According to the author, the mystical experience is defined by four characteristics: 1. Ineffability, that which defies expression and language; 2. Noetic quality, which would be the illuminative dimension (and in this, James agrees with many traditional Christian mystics that love is, in itself, a form of knowledge); 3. Transience, in the sense of mildness, tolerance, and cordial and loving openness; and 4. Passivity, or the inability to produce the experience by oneself.

For James, the mystical experience is something that takes place between affect and thought. His investigation of the "theoretic movement" of mystical states leads him to argue that such states promote the expansion of consciousness and allow "overcoming all the usual barriers between the individual and the Absolute" (James 1929, p. 329).

The great historian of mysticism, Bernard McGinn, on the other hand, considers that James' method of distinguishing between the affective state of mystical consciousness and the concepts which go beyond belief of a philosophical and theological nature sometimes grafted onto it does scarce justice to the complex interactions between experience and interpretation on the one hand, and feeling and thought on the other. Thus, according to McGinn, mysticism, for James, would be both cognitive and non-cognitive, but one can never be sure how this happens (McGinn 1992).

The Belgian philosopher, psychologist, and theologian Joseph Maréchal, a precursor of the so-called "transcendental Thomism" of Lonergan and Rahner, on the other hand, adds an additional element. In his text *À propos du sentiment de présence chez les profanes et chez les mystiques* (McGinn 1970), he reflects on mysticism, not from the perspective of union with God, but from the psychological and philosophical notions of the feeling of presence; that is, from the "a priori" conditions for the possibility that an evaluation on the existence or not of God's presence at the core of the experience is also a judgment of reality.[6]

For Maréchal, intelligence must be understood as fundamentally intuitive in its impulse and purpose. "The affirmation of reality, then, is nothing but the expression of the mind's fundamental tendency toward unification in and with the Absolute" (Cf. ibid., vol. 1, p. 101, translation ours). Mystical experience, at least at its culmination, is thus, for Maréchal, the direct, intuitive, unmediated contact in this life between intelligence and its ultimate goal, the Absolute. In his words, it is "the intuition of God as present, the feeling of the immediate presence of a Transcendent being" (Maréchal 1937, pp. 102–3). Again, we come close here to the presence of knowledge when speaking of the mystical experience.

For Maréchal, Christian mysticism would be the analogatum princeps of all that can be said about mysticism as a whole. From a comparative or descriptive point of view, Maréchal provides a broad definition: "the mystical experience in any circumstance is superior to normality: it consists of something more direct, more intimate or rarer" (Ibid., p. 288). Maréchal distinguishes between lower and higher states within mysticism, following a general Augustinian distinction of types of visions. Christian mysticism, according to Maréchal, contains three clearly defined degrees. 1: "integration of the ego and its objective content, under the preponderance of the idea of a personal God"; 2: "the transcendent revelation of God to the soul", often with the suspension of all of the latter's other activities (the level of ecstasy, a special object of much of Maréchal's research on Christianity); and 3: "a type of readjustment of the soul's faculties by which it again comes into contact with creatures under the immediate and perceptible influence of God that is present in and acts upon the soul " (Ibid.).

## 2. Mystical Experience: Loving Knowledge?

What seems to emerge from the assertions presented thus far is that the mystical experience is a knowledge experience, but in it, the presence of the other, and deep interaction and love for this experienced otherness takes precedence over mere rational knowledge.

In his book *Les Degrés du Savoir* (Maritain 1932, 1975), the great French Thomist philosopher Jacques Maritain distinguished between: 1. A negative Theologytheology of neo-platonic origin that is pure intellectual apophaticism pretending to be mystical and 2. A negative Theologytheology that springs from the naturalness of love, and is therefore the expression of the true mystical experience.[7]

Maritain defines mystical experience as "an experiential knowledge of the deep things of God", coming close to the definition attributed to Thomas Aquinas of "cognitio Dei experimentalis" (knowledge of God through experience) or even as "a possession given of experience of the Absolute".[8] According to Maritain's reading of John of the Cross, he ventures another definition, of "fruitive experience of the absolute" (Maritain 1975, pp. 1125–58, cited in de Lima Vaz 2000, p. 12, n. 4). Three things happen in the mystical union: we come to love God as he loves us because uncreated love becomes the agent of everything the soul does; the soul "gives God back to God"; and, finally, the soul comes to participate in the true life of the Trinity (Maritain 1932, pp. 373–81).

McGinn, however, aptly notes Maritain's difficulties in dealing with the use of erotic language—constantly and abundantly employed by mystics—although he almost grudgingly admits that this is the most direct analogy to mystical love. The historian observes that this is particularly puzzling for a thinker who lends affective connaturality such a central role.[9] Maritain's concept of mystical experience as loving knowledge seems to maintain a safe distance from all that is desire, eroticism, and fruition when writing on mystical experience.[10]

From a different perspective, Michel de Certeau appears as a thinker who brought a novel and original approach to the understanding of mystical experience as knowledge. The French Jesuit emphasized the way in which mystical experience, while being of profound significance in individual terms within the life of the mystic, is necessarily a social phenomenon in spite of being also an intellectual one. As a historian and psychologist, de Certeau certainly does not miss the fact that mysticism always reflects a socio-religious world that serves as its backdrop. On the other hand, de Certeau also notes that mysticism similarly affects and transforms the world through the creation of new types of discourse that is the fruit of the knowledge it brings and of the formation of new religious groups.[11]

In his classic article "Mystique," in the *Encyclopaedia Universalis*, de Certeau states: "Since the 16th and 17th centuries, mystique is no longer designated as a kind of 'wisdom' elevated to the recognition of a mystery already experienced and proclaimed in common beliefs, but an experiential knowledge that slowly separated itself from traditional theology or ecclesiastical institutions and is characterized through the awareness, acquired or received, of a gratified passivity where the 'self' is lost in God" (de Certeau 2009). De Certeau maintains that mystical experience cannot be studied in itself, but only through mystical language and through the body of the mystic (de Certeau 1982, pp. 12–15). The word "mystical" itself, although it first appeared in the writings of Dionysius Areopagite, which date back to the beginning of Christianity, has its use as a noun dated to the beginning of the 17th century in France, as Michel de Certeau's authoritative research shows (de Certeau 1963).

The path of reflection on mystical experience has undergone all these steps, which show a certain fluidity in the limits and borders of its conceptualization. Catholics and Protestants were unable to reach a consensus when the challenge was to establish the citizenship of the concept in academic theological and philosophical circles, especially Protestant ones. As McGinn demonstrates, quoting Albrecht Ritschl's definition, "[m]ysticism is therefore the practice of Neoplatonic Metaphysics and that is the theoretical norm of the alleged mystical delight in God. Consequently, universal being seen as God in which the mystic wishes to be dissolved is a fraud" (McGinn 1992, p. 267). Adolph Harnack further states

that "[m]ysticism as a rule is rationalism worked out in a fantastic way; and rationalism is mysticism faded" (Ibid., p. 268). McGinn criticizes Harnack's position, affirming that it has the vulnerability of failing to explain how the Western mystical tradition that began with Augustine relates to the Eastern tradition, whose best expression are the writings of Pseudo-Dionysius.

### 3. Mystical Experience in Contemporary Christianity

In the Second Vatican Council (1962–1965), Christian thinkers and authors, seeking to respond to secularization, did not exactly bring something new to the discussion. They were, in fact, following paths already strongly and deeply trodden by their predecessors. After the Council, Catholic Theologytheology underwent a true anthropological Copernican turn. From then on, the debates on mysticism in Catholic Theologytheology have revolved around two closely related questions: (1) Is the call to universal mystical contemplation offered to all Christians, or (2) is it a special grace available only to a chosen few?

Undeniably, the most significant Catholic contribution on mysticism in the second quarter of the last century was that of the German Jesuit Karl Rahner, named rightly, in our view, the "Doctor mysticus" of the 20th century (Cf. de Egan (1985, p. 3), quoted in McGinn (1992, p. 286, n. 107)). The dogmatic key to his notion of mysticism rests on his distinction between transcendental experience (the a priori opening of the subject to the ultimate mystery) and supernatural experience, in which divine transcendence no longer constitutes a remote and asymptotic objective of the dynamism of the human subject, but is communicated to the subject in proximity and immediacy.

Rahner insists on the reciprocal unity (but not identity) between the experience of God and the experience of the self that becomes whole in interpersonal relationships: " . . . the unity between love of God and love of neighbor is conceivable only on the assumption that the experience of God and the experience of the self are one and the same (Cf. Rahner 1975, 1976, pp. 149–65): "On both the transcendental and supernatural levels (i.e., God as question, God as answer), one must always keep in mind the important difference between experience itself and its subsequent thematization or objectification into conscious reflection as a categorical mode of thinking. Thematization can never capture the fullness of original experience, but experience cries out for thematization in order to be communicated to others."[12]

This is the reason why Rahner speaks of mysticism in two ways (McGinn 1992, pp. 286–89):

> "There is the mysticism of everyday life, the discovery of God in all things," that is, the unthematized experience of transcendence at the core of all human activity (Rahner 1984, pp. 80–84). Rahner's theology of grace suggests that this experiential substratum always operates in a mode elevated by grace, that is, that mode in which God has already responded to the call he placed in the belly of mankind, even though this may not be evident from a psychological consideration of the thematized knowledge of the acts themselves. (Ibid., pp. 75–77)

There are the "special" mystical experiences that Rahner admits can be found both within and outside Christianity. As far as the Christian faith is concerned, these experiences cannot be conceived of as constituting some intermediate state between grace and glory: they are a variety or mode of the experience of grace in the faith (Ibid., pp. 72–73). Although Rahner is firm in his opposition to any "elitist" view that claims to find in mysticism a higher form of Christian perfection beyond loving service to one's neighbor, he speaks about the special mystical experiences as a paradigmatic intensification of the experience of God that is open to all.[13]

However, Rahner himself insists that whether mystical experience is truly the pinnacle of the normal development of the subject is a question for empirical Psychologypsychology to judge, and is not within the purview of Theologytheology as such (Rahner 1984, p. 77). This is due to the Rahnerian interpretation of certain issues presented by mysticism, such as the so-called "depth experiences", "altered states of consciousness", or "experience of

suspension of faculties", which he considers to be essentially natural phenomena, potentialities of the subject, whether or not raised by grace[14]. If such experiences are judged by Psychologypsychology to be part of the normal maturation process of the subject, then mystical experience, in a special sense, whether or not thematized, will be in its entirety truly human and Christian.

For Rahner, the event of Christ is central to all mystical experience. The historical reality of Jesus, as communicated through the life of the Church, is constitutive for all forms of the salvific relationship with God. Our relationship with Jesus is unique and, in it, an immediate relationship with God is communicated through the mediation of the Savior incarnate. This is why Rahner maintains that "Christ is the 'fruitful model'" "per se" for a committed trust in the mystery of our existence.

The Rahnerian theology on mysticism, according to McGinn, offers original and profound answers to some of the basic questions in the modern discussion of this area of study and research. On the question of whether mystical experience represents a higher level beyond the ordinary life of faith (the root of many Protestant objections to mysticism), Rahner's answer is "no" theologically, and "perhaps" psychologically. On the relationship of Christian mysticism to non-Christian mysticism, he expounds and extends his famous thesis concerning the "anonymous Christian" to include the category we can name using the expression coined by McGinn, namely the "anonymous Christian mystic" (McGinn 1992, p. 288) That is, he believes that some non-Christian forms of mysticism are true expressions of the special experience of God's response given in Jesus Christ, even if not explicitly thematized, named, and known as such (Rahner n.d.).

In German-speaking theological output, next to Rahner, another great figure of theological reflection on mysticism is the Swiss theologian, Hans Urs von Balthasar. He begins by distinguishing between the objectivity of mysticism, found in the revelation of the mystery of Christ, and the subjectivity, expressed in the traditional Thomistic definition of mysticism as "cognitio Dei experimentalis" to be found in the experience of the subject.

In Balthasarian theology, the thesis of the observed similarity between Christianity and other non-Christian forms of mysticism is challenged by the theological antithesis that insists on the separation between the two. Balthasar recognizes that there are two forms of antithesis: (a) the Protestant version, which argues that mysticism is entirely unrelated to the Christian faith; (b) the more frequent version, which is the Catholic version, where only Christian mysticism is true mysticism. The synthesis, then, that can appropriate "correct" views—according to Balthasar—of mysticism within the Christian life operates according to these basic criteria: (1) the norm of the supremacy of the reciprocal love of God and neighbor; (2) the necessity of conformity to the model of Christ; and (3) the insistence on the unknowability of God in and through his manifestation in the Word made flesh. Balthasar insists that only Christian mysticism is true (McGinn 1992, p. 290).

In the postconciliar moment, therefore, Catholic thought reached some consensus regarding the concept of mysticism, namely that it is not about a special higher or elitist form of Christian perfection, but rather, a part of the demand of the life of faith itself. As for how this requirement is to be understood and, above all, lived through and practiced, there is no longer a consensus (Ibid.). There have been recent attempts by Catholic theologians to widen the field of research on Christian mysticism, attempting in various ways to reformulate traditional questions, given the post-scholastic era that was made official by the Second Vatican Council.[15]

In Spain, Juan Martin Velasco states that "relying above all on the texts of Christian mystics, although aware that our statements also find support in Muslim and Jewish mystics, we can offer, as the center and summary of mystical experience, the affirmation that in it the subject lives in the mediated immediacy of loving contact, the most intimate union with the very reality of God present in the deepest part of the subject's being" (Velasco 2009, pp. 30–31) Velasco likewise presents his own typology of mystical experience. He identifies three non-negotiable elements for mystical experience to be said to exist:

1.　　The intimate union with God as the content and goal of the experience

2.　Its condition as an immediate experience in the mediation of the soul and the traces that God's presence leaves in it

3.　Love as the way and means of union. (Ibid., p. 31)

The author argues that union is the most frequent form to express the ultimate degree of mystical experience. Other categories, namely ecstasy, contemplation, vision of God, deification, theopathic state, etc., are important, but they do not become a key category as does union.[16] Velasco is not simply concerned with Christian or even monotheistic mysticism. His books deal attentively and in depth with the question of profane, secular, or "natural" mysticism, of pluralistic mysticism that traverses or is present in elements of various traditions, and of mysticism in other great traditions, such as Buddhism and others.[17]

In his reflections on religious and mystical experience in the West and in the East, the scholar of religions and mysticism, Raimon Panikkar, also goes in that direction. He proposes pilgrimage as a symbol of life, but not as life itself, because pilgrimage must be not only exterior but also interior. Having left behind several writings on mysticism, Panikkar is one of the pioneers of the theological reflection on the dialogue on mysticism in different religious traditions (See, for instance, Panikkar 1993, 2006).

Finally, and of particular note, the Brazilian Jesuit philosopher Henrique de Lima Vaz clearly distinguishes three forms of mysticism, one of which is markedly connected to what we understand here as knowledge. Lima Vaz distinguishes those three forms without separating them rigidly, as he recognizes, with Henri Bergson, that mystical experience is an inexhaustible source of the highest ethical and religious aspirations to which a civilization can rise. His distinctions are situated within the Western tradition and are speculative mysticism, mystical mysticism, and prophetic mysticism (de Lima Vaz 2000, p. 29).

According to this author, the so-called "speculative mysticism" can be considered an extension of metaphysical experience in terms of experiential intensity. It could thus be viewed as the face of philosophical thought turned towards the mystery of Being, trying to draw its gaze into the unfathomable and ineffable depths that signal the ultimate frontier of distinctive thought and human language and logic, so that the human spirit is able to pierce the domain of the Translogic. Thus, it has tended to flourish, historically, in close connection with the great bursts of metaphysical thought which have marked the history of philosophy. In its Western version, speculative mysticism is mostly Greek, despite the vigorous growth it has experienced in Christian territories. It is situated in the noetic strand of consciousness, blossoming, as it were, at its apex, its summit. And Lima Vaz defines it very clearly: "It is, therefore, a mysticism of knowledge, and this is the original feature which distinguishes it in the history of mysticism."[18]

In attempting to evoke the reflections of the various authors studied above, we understand here by mystical experience what Philosophy and Theology agree in defining: an awareness of the divine presence, perceived immediately, in an attitude of passivity, and which is experienced before any analysis and conceptual formulation. It is the concrete experience of the human being who finds him/herself, thanks to something he/she does not control or manipulate, faced with a mystery or in closeness with a mysterious and irresistible grace. That gracious presence reveals itself as a personal alterity and acts lovingly, proposing and bringing about a communion that is impossible according to human criteria, and that can only happen graciously when coming from the other.

This experience, without failing to include all the categories that appear with greater force and consensus in the authors studied (ineffability, immediacy, and passivity, among others) is fundamentally an experience of relationship. Because of this, it presupposes the otherness and his/her difference and cannot be reduced to a symbiosis or desire for fusion, as can happen in different religious experiences. In this sense, as a relational experience, and only in the light of this first fact, can one speak of the noetic dimension of mysticism, meaning that mystical experience is an experience of knowledge. Mysticism is then, and indeed, knowledge, but a knowledge that comes from experience and where intelligence and intellect arrive afterwards, in the sense of capturing and interpreting, not the experience

abstractly speaking, but what the concrete subject feels, the sentiments of the one who is at the center of the very act of experiencing. And these feelings imply an alterity and a relationship that speaks, enters into a dialogue, and teaches. What is brought through this relationship are new illuminations and inspirations, generating new knowledge.

The path of the relationship with the Transcendent and Divine Other generates a unique form of knowledge which is, therefore, constitutive of the mystical experience itself. Regarding Christian mysticism, this other, this alterity, possesses the anthropological component at the center of its identity, since the experienced God became flesh and showed a human face. Thus, everything that comes out of mystical experience within a Christian outlook cannot deviate or even disregard that which constitutes the flesh and humanity of its neighbor. It is paradoxically in the closeness and the deepest similarity to humanity that God, according to Christian Revelation in mystical experience, will show absolutely transcendent difference and otherness. These are experienced as a powerful source of knowledge.

Our next step in this reflection will be to think about mystical experience as knowledge, with special attention paid to women. As the Italian philosopher and mysticism scholar Marco Vannini states, "as far as the Spirit is concerned, it has no gender, just as there are no distinctions of a cultural, social or environmental nature: it is universal. But for centuries the ecclesiastical institution was suspicious of women who took on a magisterial role".[19] Women have been, in the history of Christianity, and particularly in Catholicism, a continuously "suspicious" otherness.

Not having full access to education, unlike men, women found in spiritual life a pathway which brought them to reading, to reflection, and to contact with philosophical and theological thought. Nevertheless, their wisdom was not particularly integrated and accepted as part of the religious institution. The feminine body has been associated with sin, temptation, and sexuality, generating a secular discrimination within an institution unanimously ruled by men. Accused of being responsible for the entry of sin into the world, and for death as a consequence of sin, women often found in mystical experience a means to accept their body and its sensations and emotions, together with finding access to reflection and a language with which to describe their spiritual experiences (Kristeva 2014). But because of the ever-present suspicion which confronted their body and mind, the mystical experiences of many women were often placed under suspicion, and under the severe and strict vigilance of men in charge, to control and exorcise them.

Numerous and very rich mystical experiences of women who lived through very intimate and important spiritual communications and powerful philosophical intuitions and enlightenments remained ignored in a universe where the means of dissemination were in the hands of a few, and where cases such as Teresa of Avila are the exceptions that confirm the rule. Throughout the history of Christianity, women were kept at a prudent distance from the sacred and all that surrounded it, as well as from liturgy, from ritual objects and spaces, and from direct mediation with God. The same happened with their opportunity for deeper intellectual training. Between women and mystery, there has always been a distance inserted, with their experiences being recognized and legitimized rarely or with great difficulty in terms of "high" mysticism, relegating them therefore to the realm of lesser devotions.[20]

Today, the process of the emancipation of women has arrived in the Church and has questioned this state of affairs, and we can find, stronger than before, the narratives of the mystical experiences of women appearing increasingly in research and in reflections by philosophers and theologians. Among them, medieval women occupy a privileged position and are the most researched by scholarship.[21] It is again Marco Vannini who affirms that "to women we owe an essential contribution to the history of spirituality, of mysticism, especially significant for those past centuries in which women did not normally have access to education".[22] He continues, quoting Angela de Foligno and Marguerite Porete, who in spite of that, "speak not as words by women, but as magisterial words of spirituality" (McGinn 1998b).

Not only these, but also the women of the modern age and even of contemporary times can bring luminous testimonies on how their mystical experiences have been altogether a spiritual and intellectual enlightenment which left important footprints in their time and space.

## 4. Women Mystics: Love, Knowledge, and Praxis

After attempting to retrace the historical itinerary of the concept of mysticism and to discover its connections with knowledge and attempting to reflect on mysticism as knowledge by resorting to some important scholars, we will now examine how mystical experience has been a gateway to knowledge, especially for women. Given their difficulty in accessing education and reading materials, given their imposed illiteracy, many women discovered in their mystical experiences a unique way to access study, reflection, speculation, and even to build their own doctrine. The history of Catholicism, in a very special way, has some remarkable feminine figures who demonstrate this.

Important religious thinkers such as David Tracy who, in some of his recent writings, brings to light elements found in women mystics that may catch our attention: the concern for the inequality between men and women, in society and very particularly in the Church, the 'difference' of women living, thinking, and expressing the mystery of God, and the creative synthesis of which women seem especially capable in doing justice and loving service to the most vulnerable.[23]

While recognizing, for instance, the unique protagonism of Francis of Assisi not only in Christian history, but in the whole of Western history, Tracy highlights the movement created by Francis' charism, during the 12th century, where voluntary poverty configured a completely new way of being Christian. Together with Francis, a woman, his very beloved disciple and companion, Clare of Assisi, radicalized the Franciscan "privilege of poverty" for a community of contemplative life at San Damiano (Lein and Post 2010, p. 152).

In fact, Clare was merely the first in a notable tradition of women inspired by the Franciscan charism who would be creative in living out the master's deepest desire of poverty, independently from Francis and the male order.[24] A late 13th- and early 14th-century Franciscan Tertiary who has also attracted considerable attention from scholars like Tracy, but also from non-Christian scholars, such as Julia Kristeva, is Angela of Foligno (Clément and Kristeva 2001, p. 48; Kristeva 1997, op cit). In studying her experience, we can find similarities and affinities with the contemporary lay Beguine women's movement, which was significant in Europe during the late Middle Ages (McGinn 1997). One very striking point in Angela de Foligno's mystical language is her experience and reflection on the love of God as being excessive. Tracy regards himself as following Angela and recognizing her mystical knowledge, which is faithful to the best of Christian tradition, when he defines the Christian conception of love as fundamentally "excessive" in the four gospels, in Paul, in the noncanonical gospels, and in other early Christian texts. These texts, he writes, are "good news announcing and disclosing the impossible: the final truth about reality is love and God's infinite love disclosed in all excessiveness in the self-sacrificing love . . . of Jesus the Christ" (Lein and Post 2010, p. 137).

In fact, Angela de Foligno is unrivalled when it comes to excessive love. She gave up wealth, comfort, and everything she possessed to serve the poor and the outcast, even the lepers. But what is even more admirable in her is her capacity to combine this radical service to the most vulnerable with an ever-deepening interior life. "Eventually Angela became more and more committed to the interior life without ever abandoning her commitment to the poor" (Ibid.). This seems to be the very point named by Tracy here as "excessive love" from an "excessive infinite": an excessive love that overcomes all impossibility of experiencing and offering service to a God by whom she felt so deeply loved. This excess made it possible for her to combine, in her life, intense visions and extraordinary mystical phenomena, an equally intense intellectual theology of love, and untiring service to the poor and most outcast from society (Ibid.).

Angela de Foligno was able to elaborate a true "doctrine" of mystical darkness, a very original spiritual synthesis, which unites joy, the fullness of the apophatic impossibility of language, and negative experiences of deep suffering (Kristeva 2022). Her soul entered fully into a terrible journey, experiencing this excess of love together with an excess of suffering because of love.[25] Finally, Tracy concludes: "Love as infinite excess in God and finite excess in the self has never found a more brilliant exponent than Angela of Foligno" (Kristeva 2022; Lein and Post 2010, p. 153).

In addition to Angela, however, and "even more important than the birth of the great mendicant orders," there were the women mystics and theologians of the 13th and 14th centuries (Ibid, p. 152). Tracy singles out Mechthild of Magdeburg (who is herself reminiscent of one of the greatest 12th-century theologians, Hildegard of Bingen), Hadewijch of Antwerp, and Marguerite Porete. These "vernacular theologians" created theologies distinct from monastic and scholastic theologies in Latin. With Angela, they constitute, in the words of Bernard McGinn, "the four women evangelists" of the age.[26] These women follow and come close, in importance, to the paradigmatic female figures of the origins of Christianity, such as Mary Magdalene, who made the first announcement of Christ's resurrection.

The 16th century brought perhaps the greatest female protagonist in the Christian history of mysticism, not only due to her extreme sensitivity and affective depth, but for her brilliant intelligence and organizational capacity. Teresa of Avila is extraordinary in many respects and mostly because she achieves a felicitous ordinariness, which Julia Kristeva describes as a spiritual equilibrium. Teresa translates "outside time" into "ordinary time." She transforms her once-anguished life of "tortured, erotic carnality," supposedly predictable for a woman, not only into the bliss of mystical union with her divine Spouse, but also into a daily conformity to "His Majesty's" will. She eludes ailments very common in other women mystics, such as hysteria, anorexia, or epilepsy. She discovers and creates language, "transmuting her body into writing and her writing into action. Beyond being a desirous daughter and joyous wife, she becomes a highly effective *symbolic* mother: the foundress of the Discalced Carmelites and institutor of new communal norms. She forges a livable life for herself and others" (Kristeva 2014, pp. 47, 279, 303–5, 310).

Subsequent to Teresa, another woman who lived during the French 17th century as a mystic and thinker is Jeanne-Marie Bouvier de la Motte Guyon.[27] The famous Mme. Guyon was the intellectual and spiritual partner of François Fénelon, the Archbishop of Cambrai and famous writer and theologian (Tracy 2020b). Mme. Guyon was a central figure in the theological debates of 17th century France through her reflection on quietism, an extreme passivity and indifference of the soul, to the point of a peaceful indifference to one's eternal salvation, wherein she believed that one became an agent of God.

Guyon was suspected of heresy and her spirituality was denounced as unorthodox and non-authentic by great figures such as Bossuet.[28] Nevertheless, Fénelon defended her mysticism of passive contemplation as genuine, and Mme. Guyon was convinced, along with Fénelon, that her mystical experience was profound and configured by love. As Tracy notes, it was Fénelon who also understood Guyon's profound love-mysticism as a correlate to "the philosophical, active contemplation of Descartes's idea of the Infinite" (Ibid., p. 117). Fénelon's fidelity to and admiration for Guyon owed much to the way in which she was a more extraordinary spiritual writer and a more profound mystic than himself. In fact, he was the disciple, and she was the master. Furthermore, Fénelon learned from Guyon the experience of passive contemplation and the reality of pure love that he had earlier observed in his readings of St. François de Sales. Jeanne de Guyon and François Fénelon enriched each other reciprocally in intimate spiritual friendship (Ibid.). Jeanne de Guyon taught Fénelon the paths and methods of her acute mysticism, and Fénelon added philosophical arguments and theology to support her mystical experience and thinking.

Jeanne de Guyon is today considered a superior enlightening figure in Christian mystical and theological history. She was not just the inventor of a novelty, but rather a legitimate heir of the best Christian tradition, as learned from John of the Cross, Teresa

of Avila, and François de Sales, among others, which she constantly read. As frequently happens, her leadership and mystical doctrine were not understood, and she was even imprisoned, as was Fénelon. In prison, Jeanne de Guyon suffered greatly, as we know from her notebooks, found afterwards. After being released from prison, she continued to write spiritual texts and to deepen her mystical experience and doctrine.

Female spiritual and theological leadership within Christianity has never been easy. The life and writings of these female mystics have always been beset by incomprehension, and they endured very hard moments. The simple fact of being women placed them under suspicion and engendered fear, while their experiences and writings received little credibility. For some, like Marguerite Porete, this led to execution.[29] Others experienced the suffering of imprisonment and all sorts of persecution, such as Mme. Guyon, and all sorts of difficulties, such as Teresa of Avila. Others yet were marginalized by segments of the official Church and ridiculed.

Among the latter, we would be remiss not to mention, in the 20th century, the physician and doctor Adrienne von Speyr, a Protestant who converted to Catholicism and came to know intense mystical experiences (von Balthasar 2017). As with Guyon with Fénelon, von Speyr experienced a splendid intellectual, spiritual, and theological partnership with Hans Urs von Balthasar, which was not at all accepted by the religious authorities of the Jesuit order to which Balthasar belonged (von Balthasar 1991).

The importance of these mystical women is crucial for the recovery of a theology which is also imbued with Spirit and is not transformed into a dry and rigid collection of dogmatic formulae. As Balthasar wrote in a particularly famous article, there was a divorce between theology and spirituality which was very negative for both (von Balthasar 1948, n. 12). This is reaffirmed by Tracy in 2020, with an important addendum: " . . . the tragic separation of Theology and spirituality is gradually being overcome, in large part, because of the recovery of largely ignored female mystical theologians who more consistently refused any such separation" (Tracy 2020a, p. 308). Theological disfigurements of gender and form alike may be healed, as these female mystical theologians of Christian history are widely studied anew in theology schools and beyond.

In addition to being mystical and intellectual giants, all of these women practiced social justice and were devoted to the poor in one way or another. They were pioneers, as it was only in the second half of 20th century that a radical commitment to *justice* as the privileged way to live the excess of divine love took explicit shape, constructed as a preferential and structural option in the Church. We refer here to liberation theology, born in 1968 at the Latin American Bishops' Conference in Medellin, Colombia, as a result of the reception of the Vatican II Council.[30] However, some decades before, a pioneer of this option was a French philosopher and woman mystic who made a new and extraordinary synthesis which strongly supports our hypothesis of mystical experience as knowledge.[31]

## 5. Simone Weil: A Contemporary Example of Mysticism as Knowledge

Simone Weil is a philosopher who searched for truth in every moment of her life. During her very early youth, she received and believed the revelation that truth is given to those who search it with their whole heart (Weil 1966). Very accurately, David Tracy defines her as "a thinker whose very forms of thinking often act like searchlights amid our contemporary confusions; a thinker who articulated, better than anyone else of her time, why Christianity must be a mystical-political religion of and for the oppressed" (Tracy 2020c, p. 379). She believed that the rootless, spirit-bereft character of our modern bourgeois world demanded the development of a new kind of mystic-prophetic spirituality, and even a new kind of saintliness-in-the-world struggling for justice and peace (Weil 1966, p. 62).

The original and "strange" Weil recognizes herself as a Platonist. Philosophy is her field of reflection. But much like Plato, her great master, she believes and practices a philosophical thinking which is, in good part, the fruit of the contemplation of the myths.[32] Weil took odd positions about many important questions and could never be identified as a feminist.[33] And even after her mystical experience, which had a clear Christian Catholic

configuration (Weil 1966, p. 49), she remained outside the Church because she was unable to accept a few serious contradictions within the institution. According to some Weilian scholars, among them David Tracy, Marco Vannini, and Emmanuel Gabellieri (Tracy 2020c; Vannini 2014; Gabellieri 2003), this French woman—declared by Albert Camus to be the only great spirit of her century[34]—left a fundamental legacy to philosophy and theology with the new bridge she built for her time and ours between Christianity and the ancient Greeks. Weil found in the *Iliad* christic figures, like the young man who goes voluntarily to Hades, and asserted that the *Iliad*, together with the Gospels, are the only two genuine epics of the West.[35] In both the *Iliad* and the Gospels, she finds suffering and beauty, and the strange and rare covenant of these two transcendental elements inspired her entire life and thought. Having experienced pain and beauty, Weil understood and wrote about the passion of Christ, which constituted the apex of her life's final years (Weil 1966, p. 33). She produced a precious reflection with what Tracy identifies as her "tragic sensibility", by means of which Platonism avoids being just one more philosophical system and Christianity avoids Christendom (Tracy 2020c, pp. 382–83). She gives a christic configuration to the philosophical figure of *metaxu*.[36]

There is thus a unique theology in Weil's thought, according to Tracy, and as other commentators have also affirmed (Ibid.). For Weil, God is the Hidden God who has withdrawn from the world. The mystery of Christ is the maximum beauty, the passion and the cross, in a divine-human experience that causes envy within her.[37] Meanwhile, Weil's reflection on the interior life of the Trinity came close to the thoughts of theologians like Balthasar and Moltmann (von Balthasar 1980; Moltmann 1982). However, while she is struck by the dialectical hiddenness of God in the world, she is nevertheless untrammeled by theoretical questions, such as, for instance, the "sin-grace" paradigm. Her mystical experience, together with her luminous intelligence, leads her somewhere else, toward the suffering of the other, this "locus" being where she would like to be forever. This suffering of the other demands not a particular kerygmatic paradigm so much as a single-hearted attention to "the kenotic incarnation-cross of Christ, the only purely innocent, sinless human being and the one and only God" (Tracy 2020c, p. 384). In contemplating Christ's passion, the previously stoic and agnostic philosopher burns with desire to endure and live through the same. Her experience and her thought so find the synthesis in tragedy (Ibid.).

Once attuned to her Christology, this will grant her anthropology its very strong emphasis on incarnation, the cross, and her relative reticence on resurrection.[38] The same reticence or reluctance—even contradiction—can be seen in her conception of God, who is Love and Hidden as Void, and her understanding of a certain cleavage in God at creation, where "the sparks of the good are let loose in the world" and "this cleavage, disruption is transposed to the trinitarian God" (Tracy 2020c, p. 387).

While she had deep and sensitive experiences of the love of God and even unitive and loving experiences of Jesus Christ, Weil lived in permanent contradiction and radical freedom either with her spirituality or with her intellectual reflection.[39]

The integral Weil must be found not in an account of her subjectivity, nor in an account of her thought that ignores the quality of her life, but rather in the excessive desire for the Good which integrated the spiritual, the intellectual, and the practical as the dimensions of her life in just one twofold synthesis: love and truth (Tracy 2020c, p. 387).

In spite of being attracted and taken by Christ's presence, Weil was not baptized until the moment of her death (Springstead 2023)[40]. Because of this, she also never received the sacrament of Communion. She was, however, deeply in love with the Eucharist and developed a coherent eucharistic mysticism with an impressive radicalism that affected her entire life. Indeed, working in the fields on the farm of the Catholic philosopher and writer Gustave Thibon, near Marseille, Weil felt an intense spiritual comfort alongside her physical exhaustion, which she explained in clearly eucharistic terms: "The tiredness of my body and soul is transformed into food for a people that are hungry"[41]. She translated this in spiritual terms to Fr. Perrin, her Dominican confidant and friend: "My heart is transported forever, to the Blessed Sacrament exposed on the altar" (Weil 1966, p. 43). Here,

what we find in Weil is not simply adoration or spiritual love for the Eucharist, but rather a real identification built into her person and her life with this sacrament and everything it implies.

Weil was a woman who never married and whose affective life was very discrete, with no explicit signs of sexual partnerships of any kind. Neither did she experience biological motherhood. When she felt that death was approaching, she was tormented by a deep and excessive desire to give her life for the victims of war. Her choice was then not to eat more than the French were allowed within the occupied zone. Nevertheless, her true desire was to be there, suffering the same conditions as them. Faced with the inability to do so, she wrote a prayer whose radical nature has very few precedents in the history of Christian mysticism. Together with a spiritual identification with Christ's self-giving and sacrifice, it is possible to identify here a desire for the maternal gift of feeding others with one's body, configuring the whole process of bringing human life into being, something that only female bodies can experience in biological and real terms[42]:

Father, in the name of Christ, grant me this. That this body will move or not move, with perfect flexibility or rigidity, in uninterrupted accordance with your will. That this hearing, this sight, this taste, this sense of smell, this touch, may perfectly receive the exact imprint of your creation. That this intelligence, in the fullness of its sanity, will foster ideas in perfect conformity with your will. Let this sensibility feel in their greatest possible intensity and in all their purity all the shades of pain and joy. Let this love be a completely devouring flame of God's love for God. Let all this be uprooted from me, devoured by God, transformed into the substance of Christ and given to eat to sufferers who lack all forms of nourishment in soul and body.

Father, let this transformation happen now, in the name of Christ. And although I ask it with imperfect faith, answer this request as if it were pronounced with perfect faith.

Father, since you are the Good and I am only mediocre, rend this body and soul from me to make them yours, and let nothing remain of me, forever, except this rending itself, or else nothingness.[43]

Simone Weil is really a very extraordinary thinker and mystic, but also an impossible and contradictory personality. Regardless, she is a mystical thinker, possessed of enlightening theological insights; a mystic who had a mind trained according to modern illuminism, but who found mystical experience through this knowledge. Different from medieval and early modern female mystics who went from spiritual experience to coherent and deep intellectual reflection, Weil found through her speculative mysticism and her philosophical reflection the affective theological mysticism of being possessed by Christ and feeling this experience as "a smile in a beloved face" (Weil 1966, p. 76).

Weil gave the world a remarkable and unique theological understanding of the mysteries of Incarnation and the Trinity, as the central theological-spiritual clues to God's infinite and kenotic love in the mystery of incarnation and the cross. This woman, who seemed so ill-at-ease in both her religious belonging and her Jewish identity, may nevertheless be justly considered as one of the most luminous intellectual and mystical figures of contemporary times: one who lived and wrote under the force of a passionate love for the Crucifiedcrucified Christ, tormented by compassion for all human suffering. Her testimony, transiting between mystical experience and knowledge, is paradigmatic for any study on mysticism and knowledge in contemporary times.

## 6. Conclusions

We conclude these reflections on mysticism as knowledge guided by prominent scholars, but also by women mystics. The reflections of both the former and the latter have demonstrated how mystical experience is an adventure of knowledge and wisdom which can enlarge interior spaces and horizons and illuminate the paths of the self and mostly of the other.

In our postmodern society, mystical experience is particularly important because of the contribution it brings. The way of mysticism today is a countercultural way, where

humanization and the sense for life brought about by the experience of God happens in a direction contrary to society's mainstream. Its proposal for fullness, happiness, and life brings very different perspectives than those which are frequent in the media and even in churches and religions.

Knowledge brought by mystical experience introduces a path of freedom. However, this is not the freedom of immediate satisfaction of impulses and desires, nor of seduced and overflowing sensations. It is the ultimate passive experience, where pathos precedes action, where there is consciousness of one's own impotence. It is a theopathic experience which receives what is given and accepts the presence and the action of God, knowing it wasn't "produced" by oneself, but a gift and teaching from another. The teachings of this transcendent and divine otherness produces knowledge which will enlighten the subject who experiences it, and who will also communicate it to others.

In this article, we have attempted to highlight the fact that, among those who are admitted to this school of transcendental knowledge, women are especially important. For a long time, for many centuries, they have been denied access to knowledge, and confined to the loneliness, isolation, and silence of ignorance. Nevertheless, this confinement was broken when the mystical experience happened in their lives. By virtue of that gracious and excessive loving presence, they began to gain access to an intellectual mission learned from the God they experienced. They created categories and concepts, they began works of great importance, they invented languages and new alphabets, and they became experts in narratives and storytelling.

Today, the world begins to become aware of how much humankind owes to women and the feminine. There is a whole process of the emancipation of women running through society, giving them ever-increased access to a public space where they can speak and be heard. Along this liberation process, and even before the first feminist revolution started in the 20th century, there were already women mystics, creating movements such as the Beguines, thinking and writing doctrines born of their mystical experiences, such as Angela de Foligno and Jeanne Marie de Guyon, and offering the world a precious legacy of knowledge, wisdom, and practice of justice and love, such as Simone Weil did.

The mystical experience opened the hearts and minds of these women, revealing not only their affective capacity, but also their intellectual competence, which has been recognized, assimilated, and used by many other people and institutions. Because of this, they deserve to stand today among the masters and teachers who enlightened generations and opened new ways of life and growth along the history, not only of Western Christianity, but for the whole of human existence.

**Funding:** This research was funded by CITER, Universidade Católica Portuguesa.

**Institutional Review Board Statement:** Not applicable.

**Informed Consent Statement:** Not applicable.

**Data Availability Statement:** Not applicable.

**Conflicts of Interest:** The author declares no conflict of interest.

## Notes

[1]  Although we acknowledge the wonderful contributions in the mysticism of other religions—Jewish, Islamic, the millenary Far Eastern religions, and those with an African matrix—Christian mysticism is our field of expertise. By confining ourselves to it, we will be able to lend our reflection more focus and accuracy. Christian mysticism has a particular configuration because it brings the flesh close to the spirit, to the point of presenting Jesus Christ—God and man—as a pathway to human experience in the search of union with God.

[2]  In spite of taking into consideration the whole of Christianity, we will reflect mostly on Catholic female mystics.

[3]  We understand knowledge not as gnosis, the doctrine which appeared at the beginning of the Christian era, understood by its followers as mastering the secret of spiritual things, superior in nature and the purview of an elite, as the Gnostics claimed to be. Knowledge here is understood as an inspired and infused science, which enables the perception of the meaning and significance of concrete reality enlightened by the revelation of the divine.

4    Cf. Bingemer (2021). See also Bingemer et al. (2022), with a vast and diversified number of authors, from different areas of specialization, religious background, and gender. See equally our book *The Mystery and the World. Passion for God in Times of Disbelief* (Bingemer 2017).

5    For a more extensive discussion on this topic, see Bingemer et al. (2022), particularly the Introduction, pp. 11–49.

6    Cf. the famous work by Maréchal (1937).

7    Cf. McGinn's critique in McGinn (1992, pp. 306–7): McGinn states that the "rigidity of the burning Thomism of the French philosopher made it almost impossible for him to see any other philosophical system, according to his own intentions".

8    The brief definition of mysticism as "cognitio Dei experimentalis" is not really or directly found in the Summa Theologiae of Thomas Aquinas, but is provided by Gershom Scholem in his work *Major Trends in Jewish Mysticism* (Scholem 1995, p. 3). Scholem quotes the Aquinate according to the German work of Engelbert Krebs, *Grundfragen der kirchlichen Mystik dogmatischer erörtert und für das Leben gewertet* (Krebs 1921, p. 37) without confronting it with the quote of the original Aquinas (which is in STh Ij. 2quaestio 97, art. 2ad 2. Aquinas' expression is at the center of the question of whether it is a sin to tempt God and does not refer directly to mysticism.

9    (Ibid., pp. 7, 282–83). Here we agree with McGinn, cf. McGinn (1992, p. 310, n. 115), in spite of believing that this is not a problem found only in Maritain, but present in many commentators of Christian mysticism.

10    We understand here "loving knowledge" as a knowledge which happens through love experienced in the mystical union.

11    Cf. the well-known works by de Certeau (1975, 1985); as well as his major work *La Fable mystique. XVI–XVII e. siècle* (de Certeau 1982).

12    (Rahner n.d.): God and Revelation.

13    Cf. (Rahner n.d.). Sobre a intensificação dos atos religiosos, cf. (Rahner n.d.).

14    Rahner uses the German word *Versenkungserfahrungen* or *Versenkungsphanomene*, which McGinn prefers to translate as "experiences in depth" instead of "altered states of consciousness" or "experiences of suspension of faculties", as they have sometimes been translated. Cf. McGinn (1992), p. 287.

15    McGinn's work, in several volumes, is an attempt to reach this reading.

16    Cf. (ibid.) The author develops in depth the question of mystical union in his book *El fenómeno místico* (Velasco 2009), mostly in Chapter III, La estructura del fenómeno místico, Sections 5 and 6: Rasgos característicos de la experiencia mística and El núcleo originario de la fenomenología de la mística.

17    Cf. mostly the works *El fenómeno místico* (Velasco 2009), and *Mística y humanismo* (Velasco 2010). In addition to Velasco, there are other scholars who adopt this interreligious approach.

18    (Ibid., p. 30). Our translation. Let us remark, nevertheless, that he calls it a mysticism of knowledge, but doesn't affirm it like a mysticism as knowledge.

19    Cf. the interview with Marco Vannini in Vannini (2011, 2013).

20    On this topic, see (Bingemer 2018).

21    In this regard, see the works by Bernard McGinn (McGinn 1998b).

22    Vannini (2011), interview quoted above.

23    For this particular attention to women mystics, see Lein and Post (2010, pp. 131–62). See also Tracy (2020b, pp. 93–128).

24    In this regard, see the excellent doctoral thesis by Delir Brunelli (1998). See also Leonardo Boff (2006), where the author, similarly to David Tracy, places much value on the importance of Clare in life and work of Francis. See also the articles Boff wrote on Clare herself: Boff (2011, 2012).

25    (Ibid.). See also the comment by Julia Kristeva on that same woman mystic about the excess of love which reduces to all language immobility and silence (la langue coupée) (Kristeva 2022).

26    (McGinn 1998a, pp. 141–42), quoted by Tracy, 'God as Infinite Love', p. 153.

27    On Mme. Guyon, see (Millot 2006).

28    Jacques-Bénigne Lignel Bossuet (b. 27 September 1627, d. 12 April 1704) was a French bishop and theologian, renowned for his sermons and other addresses. He is considered by many one of the most brilliant orators of all time and a masterly French stylist. Court priest to Louis XIV of France, Bossuet was a strong advocate of political absolutism and the divine right of kings. He argued that government was divinely ordained and that kings received sovereign power from God. He was also an important courtier and politician. For more, cf. https://en.wikipedia.org/wiki/Jacques-B%C3%A9nigne_Bossuet (accessed on 26 November 2022).

29    See (Porete 2011). On Porete, see the work by Ceci Baptista Mariani (2008).

30    On the roots of Liberation Theology, see (Bingemer 2016).

31    On Simone Weil as a pioneer for Liberation Theology, see our article "Affliction and option for the poor", in Rozelle and Stone (2009).

32    See, for instance, the first academical work she did for Alain at the Sorbonne, about a Grimm's tale: https://www.worldoftales.com/fairy_tales/Brothers_Grimm/Margaret_Hunt/The_Six_Weilans.html#gsc.tab=0 (accessed on 24 November 2022). On this

topic, see our article: Os seis cisnes e a menina. Algumas reflexões sobre a salvação segundo Weil, in https://revistas.pucsp.br/index.php/teoliteraria/article/view/22940 (accessed on 24 November 2022).

33 Among those, there is her attitude in front of Judaism, about the Church, and many other points. See about it all her writings to her friend Fr. Joseph Marie Perrin in *Attente de Dieu*, or to Fr. Couturier in *Lettre à un Religieux*, digital version, 1951.

34 In this regard, see https://www.simoneweil.com.br/depoimentos/camus, a letter addressed to Weil's mother, where the Nobel prize writer Albert Camus underlines the word "only" (seul) (accessed on 2 January 2023).

35 Cf. our book "*Simone Weil la forza e la debolezza dell'amore*" (Bingemer 2007).

36 Emmanuel Gabellieri, Être et don, op. cit.

37 (Weil 1966, p. 49), Autobiographie Spirituelle.

38 In Weil (1954, p. 58): 'Hitler could die and return to life again fifty times, but I should still not look upon him as the Son of God. And if the Gospel omitted all mention of Christ's resurrection, faith would be easier for me. The Cross by itself suffices me . . . the proof, the really miraculous thing, is the perfect beauty of the accounts of the Passion, together with certain glowing words of Isaiah's: "He was oppressed, and he was afflicted, yet he opened not his mouth . . . " and of St. Paul's: "Who, being in the form of God, thought it not robbery to be equal with God; but made himself of no reputation . . . and became obedient unto death, even the death of the cross." . . . That is what compels me to believe.'

39 On Weil's mystical experience, see (Bingemer 2015).

40 Cf. the excellent analysis of this event in Nicola et al. (2005). See also Springstead (2023).

41 As stated in the letter to Simone Pétrement, in Perrin and Thibon (1967, p. 138).

42 On this theme, see my article (Bingemer 2014).

43 Weil (1970, pp. 243–44). Originally in Weil (1950).

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
