# Peer review of "Mystical Experience: Women’s Pathway to Knowledge"

_religions, doi:10.3390/rel14020230_

Round 1

Reviewer 1 Report

Unfortunately, my view is that this manuscript is not ready for publication in its current iteration. This is mainly because of structural flaws stemming from its conception. The attempted research is too broad to be brought to successful fruition. The link between mysticism and knowledge, without much qualification, is an area of theological and philosophical studies that goes far beyond the scope demonstrated in this article. Equally, insufficient awareness is shown of studies of women mystics across cultures, religions, and diverse chronological and geographical frameworks. Ultimately, the final section of the article, focused on Simone Weil, seems like an arbitrary choice. Overall, the article would have needed a much narrower, clearer approach to meet minimum expectations for consideration of publication.

Whilst there are sections and passages that display merit and potential, for instance in the commentary to Weil's mystical work towards the end, the narrative of this article is generally weak: there are significant gaps in the transitions between sections, and the conclusions reflect an approach that is too ambitious to be realistic. As noted earlier, the links between mysticism, knowledge, and finally gender, are merely scratching the surface here.

My advice would be to focus exclusively on elements of Weil's work, and to consider a clearer approach to the subject, allowing for more specific discussion.

Equally, the article reads like a rather mechanistic translation from a foreign language. Awkward constructions, linguistic, syntactic and style errors are too many to be counted. This will need extensive editing of English language, to a degree that once again seems unrealistic.

Author Response

Thank you for your comments and suggestions, which I appreciate.

It is true that the article doesn´t cover everything on the theme (mysticism and knowledge) but tries to make a focus on specific articulations in philosophy and theology.  And mostly in Cristian mysticism .  I tried to emphasize it more in the correction.

Also the glorious history of women mystics is not entirely covered.  It was impossible to do so in an article and even in a whole book.  I selected some to demonstrate what I was trying to show with the text: that mystical life has ben for many women an opportunity for knowledge and that shaped differently their legacy.

About Weil being arbitrary, I tried to show Weil as a prototypical example of what was said before, with the differente that she began on the other end.  She is not a mystical who has access to intellectual life, but an intellectual who has a mystical experience and life and that impacts in all her life and work.  But I agree with you that it would be much more to be said about that extraordinary woman.  I hope to offer not so far a reflection only about Weil.

The translation was revised by a professional and I think is much better now.

Thank you again.

Author Response

Thank you for remarking the need for English revision.  At this moment a specialist English reviewer is finishing to work on the article and, in my opinion, is making a splendid work.

Thank you so much

Reviewer 3 Report

Review

“Mystical experience: women gateway to knowledge”  

In general.

The manuscript is well written, clear in structure and well documented. The title covers the content. The conclusion is supported by the reasoning. The topic suits the journal. It offers a review of mystical experience and the contribution of women to mysticism. The text has mainly the character of a review and makes me sometimes wonder what position the authors themselves take. A bit more critique (for or against particular interpretations) could improve the paper. Both as review and reflection, the text contributes to our comprehension of the role of women in (Christian) mysticism and is worthwhile publishing. I like to congratulate the author(s) with this accomplishment. It was a pleasure to read the manuscript. The work must have been extensive and I suppose it results from a PhD thesis.  

For the authors, I have the following questions and comments:

1.     I think the authors should somewhere in the introduction delineate their account to Christian mysticism as other kinds of mysticism (Jewish, Islamic, Indian) are not discussed. 

2.     In the review and discussion of female mystics, I miss the (supposed) female gospel writer Mary Magdalene. In feminist studies, the gospel of Mary it is even pointed out as the one grounding the four other gospels out of her experience with (the resurrection of) Christ. Also, the form (“Dialogue of the savior”) seems to point to knowledge (dialogue) and mysticism (e.g. the direct experience of Christ). What is the opinion of the author(s) on this? And, could they add a paragraph on Mary Magdalene when mentioning female evangelists (line 461).

3.     The authors claim a mystical experience is a kind of knowledge. It reminds me of “gnosis”, i.e. a seeing true/deeper than reality. As the gnosis is not necessarily Christian, I would like to comment the authors on this. So, how is their analysis of the mystical experience related to the ancient tradition of gnosis?

4.     From a philosophical point of view, knowledge is a justified true belief. Therefore, I like to ask what kind of knowledge is meant when a mystical experience is called a kind of knowledge and what is its feminine aspect? So, I would like to invite the author to specify the definition of knowledge somewhat more with regard to mysticism in general and with regard to feminism in particular (when Simone Weil is discussed).

5.     The structure of knowledge is intentional, i.e. knowledge is about something. If a mystical experience is a kind of knowledge, what is according to the author(s) its object? God or love?

6.     Also, the concept “experience” is discussed. It needs some clarification. Sense experience is not involved here, I suppose. How do the authors (in the end) define the experience involved in mysticism? Love?

In particular:

Line 19: “Mystical experience is an experience of love and affective union.” Please, do not define an experience by an experience… Anyway, not a good sentence to start with, as the different interpretations of the concept are still to be discussed in the paragraph: “Mystical experience: discussions about the concept.” 

Please correct typos in line 41 (forward), line 45 (always an always?) and 479-480 (why words underlined?).

Line 64: “the master of her life… “ What is meant by the master? Master in sense of teacher or otherwise?

Line 85 “…a knowledge over all knowledge”Wondering about the definition of knowledge. Do the authors imply knowledge of God or otherwise? 

Line 90 “ …we have to try to encircle the concept.” Mystical experience of knowledge? 

Line 98 “ineffability, that which defies expression and language” versus Wittgenstein: What we cannot speak about we must pass over in silence.

Line 127. “Again we approach here from knowledge when speaking of mystical experience.” Sentence not clear. Please explain.

Line 143: “…a knowledge experience”. What is meant here? A cognitive experience? A knowledge being transmitted? Please explain.

Line 162: “…loving knowledge”. What is meant here. Please explain this kind of knowledge.

Line 197: “…In Vatican Council II (1962-1965), Catholic or Protestant authors,” Protestant author? Please explain as Protestants do not participate in a Vatican Council.

Line 347: “…a knowledge that comes from experience” . I suppose it is not empirical knowledge being meant here. How would the authors call this kind of knowledge? Gnosis?

Line 402: I wonder if the authors by this paragraph do suggest love is the female aspect of mysticism? (Which regular interpretations often suggest). Is this a true or a rather male interpretation of the mystical experience?

Line 639: Also, with respect to Simone Weil, I wonder what kind of knowledge is being meant here? I think Weil’s life is a form of modern devotion by following the Christ. So, the knowledge acquired is a life (time) experience. Could this be intensified or somewhat modified by her mystical moments?

Line 713: , in stead of :

My advice: accept the paper in its current form with some reflection on the remarks being made in this review added.  

Author Response

Thank you so much for your comments and suggestions.  They enrich our work.

About the other traditions´mysticism I tried to insert it in a note, once the article focus on Christian mysticism.

The case of Mary Magdalene is complex, because she is not considered as a mystic, but rather as a disciple as the other 12, and the first witness of Resurrection, briefly as a source, a founder of Christianity. Your comment acknowledge that. Mary Magdalene was, yes, a source for Christian mysticism. The women evangelist, as Mc Ginn call them, are much further and passed the crible of Constantinian shift and lived in the middle of Christendom, in Middle Ages.  That is why I didn´t include Mary Magdalene with them.  Anyway, I think this one would merit a whole issue on her.

Also I tried to welcome and integrate your precious suggestions.

Thank you again

Reviewer 4 Report

I recommend publishing this paper with a few changes.

My main concern is that the article is may be too theological for a general religious studies journal. “Theology” is included in the key words, but I think that it should at least be made more explicit in the text. Also except for the discussion of William James and one paragraph on Protestantism, it is exclusively about Catholic thinkers, and that too should be made explicit – e.g., on page 3, line 94 it should state that the thinkers considered are all Catholic.

(1) Page 1, line 19: the characterization of “mystical experience” doesn’t even cover all of Christian mysticism, let alone all of the world’s mysticism, and should be qualified.

(2) Page 5, line181: Actually, Dionysius the Pseudo-Areopagite only used the word “mystical.” “Mysticism” and “mystic” are modern inventions, as de Certeau shows.

(3) Page 10, line 380: “truly graced by God.” That would be fine in a Christian theology journal, but it has to qualified here since not all mystics see it that way and also whether the experiences are in fact veridical is an issue. Especially if this paper is for the special issue on Philosophy of Mysticism, some acknowledgment of the philosophical issues should be made. Same with page 14, line 514.

(4) Page 16, line 610: what is this for?

(5) Wording: some errors or at least awkward phrases should be changed. For example, the title: “women gateway to knowledge”.

Page 1, line 45: “becoming always an always more ...” (Becoming an ever-more ...)? 

Page 5, line 180: “body of the mystic”?

Page 8, line 280: maybe “genuine” or “authentic” may be better than “true” here.

Page 8, line 311: “interreligious mysticism” is an odd phrase. Maybe something about “interreligious dialogue” or “mysticism in different traditions”.

Page 9, line 319: “domain of the Translogic.” Is that a phrase from Lima Vaz (I am not familiar with his work).

Page 11, line 411: “thinker,” not “thinkers”

Page 13, lines 479-480: I don’t see why the underlining is there for.

Page 13, line 483: identify Bossuert.

Page 13, note 74: “About Porete, see the work ...”

Page 16, line 583 and also note 95: change “SW” to Weil 

Page 17, line 648: either “postmodern” or “post-modern”

Page 17, line 654: delete “in”

Page 18, line 671: “the feminine”

Copy editor: sometimes there is a space between a period and a footnote number in the text and sometimes there isn’t (e.g., page 2, lines 56 and 59). Maybe the Author can correct that. Otherwise it is an issue for the copy editor. Also page 11, line 413: delete space before “:”.

Author Response

Thank you very much for your comments and corrections, which I received and accepted.  You are right that Theology imposes over Philosophy in the article. Perhaps because, I the author, am a theologian?  Anyway I tried to soften this theological predominance.  The same with the catholic prevalence.  It is true that I focus more on Catholic mystical tradition, but I tried to highlight more the ecumenical perspective.

Your specific suggestions, I think I covered all of them.

Thank you again.